# Predicting spatio-temporal wildfire propagation with dynamic firebreaks

Jiahe Zheng<sup>1</sup>, Zhengsen Xu<sup>2</sup>, Rossella Arcucci<sup>3,4</sup>, Sandy P. Harrison<sup>5,6</sup>, Lincoln Linlin Xu<sup>2</sup>, and Sibo Cheng\*<sup>7</sup>

**Correspondence:** Sibo Cheng\* (sibo.cheng@enpc.fr)

Abstract. Wildfire management strategies increasingly demand accurate predictive models that integrate real-time intervention measures. Despite advances in machine learning (ML) for wildfire modelling, existing approaches largely overlook the role of firebreak placement. In this work, we present the first deep learning-based predictive model for simulating spatio-temporal wildfire propagation with dynamic firebreaks. Utilizing a Convolutional Long Short-Term Memory (ConvLSTM) architecture, the model captures both the spatial and temporal complexities of wildfire spread while incorporating data on firebreak positioning and effectiveness. Our training dataset, derived from Cellular Automata (CA) simulations, integrates key geophysical parameters and human intervention strategies, including temporary and permanent firebreaks. Model validation across three major wildfire events in California demonstrates robust performance, with significant accuracy gains in scenarios involving strategic firebreak placement. This integration of movable firebreak placement into a wildfire spread model provides a tool for improving real-time wildfire management efforts.

<sup>&</sup>lt;sup>1</sup>Department of Mathematics, Imperial College London, London SW7 2AZ, UK

<sup>&</sup>lt;sup>2</sup>Schulich School of Engineering, Department of Geomatics Engineering, University of Calgary, 2500 University Dr NW, Calgary, T2N1N4, Alberta, Canada

<sup>&</sup>lt;sup>3</sup>Data Science Institute, Department of Computing, Imperial College London, London SW7 2BX, UK

<sup>&</sup>lt;sup>4</sup>Department of Earth Science & Engineering, Imperial College London, London SW7 2BX, UK

<sup>&</sup>lt;sup>5</sup>Leverhulme Centre for Wildfires, Environment, and Society, London SW7 2AZ, UK

<sup>&</sup>lt;sup>6</sup>Geography & Environmental Science, University of Reading, Reading RG6 6EU, UK

<sup>&</sup>lt;sup>7</sup>CEREA, ENPC, EDF R&D, Institut Polytechnique de Paris, France

# **Main Notations**

# **Main Notations**

| Notation               | Description                                                                                           |  |  |
|------------------------|-------------------------------------------------------------------------------------------------------|--|--|
| Cellular Automo        | ata Simulator                                                                                         |  |  |
| $P_{ m burn}$          | Probability that a cell that can be burned but not ignited (State 2) ignites if a neighbouring cell   |  |  |
|                        | is burning.                                                                                           |  |  |
| $R_{\rm burned\_down}$ | Probability that a burning cell (State 3) transitions to a burned-down state (State 4).               |  |  |
| $R_{ m pfb}$           | Suppression rate for cells in a permanent firebreak (State 5).                                        |  |  |
| $P_{\rm pfb\_burn}$    | Probability that a cell in a permanent firebreak burns, calculated as $(1-R_{\rm pfb})P_{\rm burn}$ . |  |  |
| $p_h$                  | Base burning probability.                                                                             |  |  |
| $p_{ m veg}$           | Factor accounting for local vegetation density.                                                       |  |  |
| $p_{ m den}$           | Factor accounting for canopy cover.                                                                   |  |  |
| $p_s$                  | Slope effect on fire spread, modelled as $p_s = \exp(a\theta_s)$ .                                    |  |  |
| $\theta_s$             | Slope angle between adjacent or diagonal cells.                                                       |  |  |
| $p_w$                  | Wind effect on fire spread, modelled as $p_w = \exp(c_1 V_w) f_t$ .                                   |  |  |
| $V_w$                  | Wind speed in meters per second.                                                                      |  |  |
| $\theta_w$             | Angle between wind direction and potential fire spread direction.                                     |  |  |
| $c_1, c_2$             | Tunable coefficients for wind effect.                                                                 |  |  |
| ConvLSTM Mod           | lel                                                                                                   |  |  |
| $f^{	ext{ConvLSTM}}$   | The ConvLSTM predictive model.                                                                        |  |  |
| W                      | Learnable weight matrix of filter parameters.                                                         |  |  |
| $\sigma$               | Sigmoid activation function.                                                                          |  |  |
| $f_t, i_t, o_t$        | Activations of the LSTM forget, input, and output gates, respectively.                                |  |  |
| $\tilde{C}_t$          | Candidate cell state value at time $t$ .                                                              |  |  |
| $C_t$                  | Actual cell state at time $t$ .                                                                       |  |  |
| $N \times M$           | Dimensions of the field being processed, where N is the number of rows and M is the number            |  |  |
|                        | of columns.                                                                                           |  |  |
| b                      | Batch size for model training and prediction.                                                         |  |  |
| S                      | Set of possible pixel states or classes.                                                              |  |  |
| $\mathbf{x}_t$         | Burnt area at time $t$ of dimension N $\times$ M, generated by the CA simulator.                      |  |  |
| $\mathbf{z}_t$         | Model output at time $t$ , representing a hidden state feature matrix that predicts the next fram     |  |  |
|                        | of wildfire progression in an $N \times M$ grid format.                                               |  |  |

15

#### 1 Introduction

In recent years, extreme fires have become more frequent, driven by ongoing climate change (UNEP, 2022; Cunningham et al., 2024). There is a growing literature on management strategies to prevent or minimise fires (Spadoni et al., 2023; Oliveras Menor et al., 2025; Neidermeier et al., 2023). However, it is also important to develop effective strategies to reduce the impact of fires when they occur. One measure that is frequently used is the creation of temporary firebreaks, through the use of fire retardants (Gimenez et al., 2004; Altamimi et al., 2022; Goldberg, 2022). The effectiveness of such a measure is substantially affected by uncertainties in the propagation of an individual fire caused by short-term variability in both meteorological conditions and fire behaviour (Hilton et al., 2015). Thus, the accurate prediction of fire spread in near-real time and how this would be affected by potential management actions would be a useful tool for proactive fire reduction.

In recent years, machine learning (ML) techniques have gained significant attention in the analysis of dynamic systems, particularly in wildfire prediction (Jain et al., 2020; Xu et al., 2024). These techniques are recognized as invaluable tools for spatio-temporal forecasting due to their ability to efficiently process large datasets and uncover complex patterns within historical data. Various approaches have been explored in wildfire modelling, such as convolutional autoencoders (Huot et al., 2022; Cheng et al., 2022a), recurrent neural networks (RNNs) (Natekar et al., 2021; Cheng et al., 2022b), and, more recently, transformer-based models (Miao et al., 2023; Masrur et al., 2024). Given the inherent temporal dynamics of wildfire spread, Long Short-Term Memory (LSTM) networks — an advanced form of RNN designed to capture time-sequential patterns — have been widely used to model the progression of fire over time (Cheng et al., 2022b; Liu et al., 2022; Liang et al., 2019; Natekar et al., 2021). In particluar, Kondylatos et al. (2022) have shown that deep learning (DL) techniques, including LSTM and Convolutional Long Short-Term Memory (ConvLSTM), are more effective than shallow ML methods like Random Forest and XGBoost in predicting wildfires in the Mediterranean region. While incorporating advanced RNN architectures significantly enhances predictive accuracy in wildfire modelling, the integration of human actions in these models requires further exploration.

Despite significant advances in ML models and the availability of numerous open-access benchmarking datasets (Huot et al., 2022; Kondylatos et al., 2023; Singla et al., 2020) for performance evaluation, no existing ML predictive or surrogate model explicitly addresses the impact of real-time firebreak placement. Mutthulakshmi et al. (2020) outline two main firefighting strategies: temporary holding firebreaks (e.g., water or chemical firebreaks deployed by aircraft) and permanent firebreaks (e.g., cleared or fuel-poor areas constructed using machinery) and emphasizes that the strategic positioning and selection of firebreaks can optimize the management of the burning area. Experimental findings from Alexandridis et al. (2011) show that the number of burning cells can be decreased by 56 % with adequate resources. However, simulating fire propagation with suppression, given the complexity of geophysical parameters, presents a substantial computational challenge. The high computational costs often prevent real-time prediction, which is essential for timely intervention. The Mutthulakshmi et al. (2020) fire-suppression model using Cellular Automata (CA), for example, simulates fire spread with human interventions but has significant memory and time demands—especially for large areas—which limits its usefulness. Similarly, the Discrete Event System Specification (DEVS) (Ntaimo et al., 2004) struggles to meet real-time requirements when updating fire parameters

https://doi.org/10.5194/egusphere-2025-4007 Preprint. Discussion started: 12 November 2025

© Author(s) 2025. CC BY 4.0 License.

based on previous states. Recent work (e.g., Murray et al. (2024); Altamimi et al. (2022)) has used reinforcement learning for optimal firebreak placement. Pan et al. (2024) presents a framework that integrates convex neural network-based fire spread prediction with optimization methods to coordinate drone swarms for active wildfire suppression. Meng et al. (2023) introduces a 3D visualization approach based on CA for simulating fire spread with the inclusion of temporal firebreaks. However, the forward predictive models in these approaches are often simulation-based, which limits them to somewhat simplified wildfire scenarios due to computational costs.

In this paper, we develop a computationally efficient fire propagation surrogate model that accounts for both permanent and temporary firebreaks. Using a CA framework, we simulate fire dynamics under various environmental conditions across three wildfire-affected locations. The model incorporates firebreak data along with local geophysical parameters such as vegetation, slope, and wind speed. We then train a DL surrogate model based on the ConvLSTM algorithm to predict fire spread. The model is validated using test data from the CA simulations.

#### 2 Data and Methods

70

Maps for each of the study areas were processed using remote sensing images of the Moderate Resolution Imaging Spectroradiometer (MODIS) and Visible Infrared Imaging Radiometer Suite (VIIRS) satellite which are available at the Interagency Fuel Treatment Decision Support System (IFTDSS) (Drury et al., 2016). We used three wildfire events in California: the 'Chimney' fire in 2016 (Chimney 2016) <sup>1</sup>, the 'Ferguson' fire in 2018 (Ferguson 2018) <sup>2</sup>, and the 'Bear' fire in 2020 (Bear 2020) <sup>3</sup> (Fig. 1).

#### 2.1 Cellular Automata Fire Simulation

We build upon the CA model that was validated through the simulation of the Spetses wildfire in Greece in 1990 (Alexandridis et al., 2008), to generate the data sets used for training and testing our DL surrogate model. This CA model utilizes square meshes to simulate the stochastic spatial spread of wildfires in a computationally efficient way. By dividing a two-dimensional terrain into  $3 \times 3$  grids, the model allows fire propagation in eight possible directions determined by evaluating the state of a central cell based on the states of its neighbouring cells (Alexandridis et al., 2011). The accuracy of the model when fire suppression strategies were included was validated by using the 2014 Dumai forest fire over a 14-day period (Mutthulakshmi et al., 2020).

Our CA model incorporates local environmental parameters such as forest information, vegetation density, slope, and meteorological data such as wind speed and wind direction to simulate fire dynamics. Training datasets were derived from three recent fires in California. Various firebreak placement scenarios were evaluated to assess their impact on fire spread. Specifically, we implemented new states in the CA model to represent permanent and temporary firebreaks. For temporary firebreaks, we developed an approach that encodes their remaining duration, allowing the model to track their effectiveness over time. The states of each cell within the grid evolve through discrete time steps as follows:

<sup>&</sup>lt;sup>1</sup>https://wildfiretoday.com/tag/chimney-fire/

<sup>&</sup>lt;sup>2</sup>https://wildfiretoday.com/tag/ferguson-fire/

<sup>3</sup>https://wildfiretoday.com/tag/bear-fire/

**Figure 1.** The Ferguson 2018 fire landscape presents information in slope, vegetation density, and canopy cover (Fig.(a), (b), (c), (d)). Similarly, the Bear 2020 fire depict distinct topographical and ecological features, including slope, vegetation density, and canopy distribution (Fig.(e), (f), (g), (h)). Fig (a,e) are from © Google wildfire product.

- State 1: The cell contains no fuel and cannot burn.
- State 2: The cell contains fuel but has not yet ignited.
- State 3: The cell contains fuel and is actively burning.
- State 4: The cell has burned out and can no longer ignite.
- State 5: The cell is part of a permanent firebreak.

States 15 → 6: The cell is part of a temporary firebreak that will transit from state 15 to state 6 over 10 time steps, before reverting to its original state.

The transition between CA states evolves over time (Fig. 2), and the cells that are either non-burnable (State 1) or have already burned (State 4) do not change state. Burnable cells (State 2) have a probability  $P_{\text{burn}}$  of igniting if one or more of their neighbouring cells are burning. Fire spreads to adjacent cells through stochastic transitions from State 2 to State 3, with the ignition probability determined by a probabilistic rule defined in Equation 1:

Figure 2. State transition pipeline of CA when the neighbouring cell is burning

$$P_{\text{burn}} = p_h (1 + p_{\text{veg}}) (1 + p_{\text{den}}) p_s p_w, \tag{1}$$

where  $p_h$  denotes the base burning probability, while  $p_{\text{veg}}$ ,  $p_{\text{den}}$ ,  $p_s$ , and  $p_w$  correspond to local environmental factors such as vegetation density, canopy cover, slope, wind speed and wind direction, respectively. These parameters are sourced from the IFTDSS (Drury et al., 2016). The influence of slope on fire spread is modelled following Weise and Biging (1997), with the slope effect  $p_s$  given as:

$$p_s = \exp\left(a\theta_s\right) \tag{2}$$

where a is a dimensionless constant, and the slope angle  $\theta_s$  is calculated using the following expressions:

$$\theta_s = \begin{cases} \tan^{-1}\left(\frac{E_1 - E_2}{l}\right), & \text{for adjacent cells} \\ \tan^{-1}\left(\frac{E_1 - E_2}{\sqrt{2l}}\right), & \text{for diagonal cells} \end{cases}$$
 (3)

Here,  $E_1$  and  $E_2$  denote the elevations of the respective cells, and l represents the cell length. The wind effect is modelled following the method proposed in Alexandridis et al. (2008), where:

$$p_w = \exp(c_1 V_w) f_t, \quad f_t = \exp(V_w c_2 (\cos(\theta_w) - 1)).$$
 (4)

 $V_w$  represents the wind speed in meters per second, and  $\theta_w$  is the angle between the wind direction and the potential fire spread direction (Equation (4)). The coefficients  $c_1$  and  $c_2$  are tunable parameters that modulate the wind's effect on fire propagation (Alexandridis et al., 2008). Wind data, including both speed and direction, were taken from Hersbach et al. (2017). Wind conditions were assumed to be spatially constant over a  $27 \, \mathrm{km} \times 27 \, \mathrm{km}$  grid, and the burned area state were resized to  $128 \times 128 \, \mathrm{pixels}$ . Each CA simulation time step corresponds to approximately 6 hours (Cheng et al., 2022b).

The operational parameters  $p_h$ , a,  $c_1$ , and  $c_2$  significantly influence fire spread predictions. In Alexandridis et al. (2008), these values are calibrated as follows:

105 
$$p_h = 0.58$$
,  $a = 0.078$ ,  $c_1 = 0.045$ ,  $c_2 = 0.131$ 

These values are derived by minimizing a cost function that fits observed fire spread data from specific wildfire events, and are used as initial values in the parameter identification process. Finally, burning cells (State 3) transit to a burned state (State 4) with a fixed probability  $R_{\rm burned\_down} = 0.4$  throughout the entire simulation.

Our CA model incorporates both temporary and permanent firebreaks. Temporary firebreaks are flexible in their placement and provide complete fire suppression for a limited duration of 10 time steps, equivalent to approximately 3 days (with each time step representing 6 hours in real time). In contrast, permanent firebreaks require a minimum distance of 1 km (equivalent to 5 pixels in the CA model) from the fire front and offer a suppression rate of about 90 % (Plucinski et al., 2007). Both types of firebreaks are subject to resource constraints, limiting their maximum extent (in our model that is limited to 50 pixels for each type of firebreak).

For cells affected by a permanent firebreak (State 5), the suppression rate ( $R_{\rm pfb}$ ) is 90%. The probability that a cell under the influence of a permanent firebreak will still burn,  $P_{\rm pfb\_burn}$ , is given by:

$$P_{\text{pfb\_burn}} = (1 - R_{\text{pfb}})P_{\text{burn}} \tag{5}$$

where  $P_{\rm burn}$  is the standard burning probability, calculated using Equation (1).

Temporary firebreaks (States 15 to 6) are assumed to provide 100 % fire suppression for an effective period of approximately 3 days. In the CA simulator, these temporary firebreaks remain effective for 10 time steps before reverting to their original state, allowing fire spread to resume if conditions permit.

The training data set generated by the CA simulator (Fig. 3) uses data from each landscape: 'Chimney 2016', 'Ferguson 2018', and 'Bear 2020'. Each dataset was generated with random wind directions, randomly positioned fire ignition field, three temporal firebreaks and one permanent firebreak positioned around the ignition field and the CA model simulates fire propagation for 26 time steps, approximately 7 days in real time. Firefighting strategies typically involve placing firebreaks along the active fire front to slow or stop the spread. However, due to the unpredictable and often rapid progression of wildfires, it is not always possible to deploy firebreaks in optimal locations. In our study, we used randomly positioned firebreaks to evaluate whether the DL model can still accurately predict fire propagation under less controlled conditions.

**Figure 3.** The fire propagation simulation of 16 time steps using CA on C ('Chimney 2016'), F ('Ferguson 2018') and B ('Bear 2020') with three temporary firebreaks (purple) and one permanent firebreak (blue).

#### 2.2 ConvLSTM Model

130 We construct a DL surrogate model trained exclusively on datasets generated by the CA simulations. By learning from the CA model's outputs, the DL model captures the underlying spatio-temporal dynamics and serves as a data-driven approximation of the CA-based wildfire propagation process with significantly improved computational efficiency by leveraging GPU acceleration.

160

RNNs, a subclass of DL models, are highly suitable for capturing complex temporal patterns. However, encoding inputs into low-dimensional representations could distort essential spatial details. The ConvLSTM architecture, introduced by Shi et al. (2015), addresses this by integrating Convolutional Neural Network and LSTM components into a unified model, and thus effectively retaining spatial information while simultaneously modelling temporal dynamics. This design optimizes computational efficiency by leveraging parameter sharing and sparse connectivity.

In the ConvLSTM framework, the input, forget, and output gates, as well as the cell states, are represented as 3-dimensional tensors. The state update mechanism employs convolution operations, thereby maintaining the spatial structure of the data. The equations governing these processes are:

$$i_{t} = \sigma(W_{xi} \otimes \mathbf{x}_{t} + W_{hi} \otimes \mathbf{z}_{t} + b_{i}),$$

$$f_{t} = \sigma(W_{xf} \otimes \mathbf{x}_{t} + W_{hf} \otimes \mathbf{z}_{t} + b_{f}),$$

$$o_{t} = \sigma(W_{xo} \otimes \mathbf{x}_{t} + W_{ho} \otimes \mathbf{z}_{t} + b_{o}),$$

$$\tilde{C}_{t} = \tanh(W_{xs} \otimes \mathbf{x}_{t} + W_{hs} \otimes \mathbf{z}_{t} + b_{s}),$$

$$C_{t+1} = f_{t} \odot C_{t} + i_{t} \odot \tilde{C}_{t},$$

$$\mathbf{z}_{t+1} = o_{t} \odot \tanh(C_{t+1}),$$

$$(6)$$

where  $\mathbf{x}_t \in \mathbb{R}^{N \times M}$  denotes burnt area at time t that is generated by the CA simulator and used as the model input. This image represents a wildfire-affected area in an  $N \times M$  grid format, enhanced with data from human interactions. Each pixel in  $\mathbf{x}_t$  can assume any value from the set S.  $\mathbf{z}_{t+1} \in \mathbb{R}^{N \times M}$  represents the model's output at time t+1, serving as a hidden state feature matrix that predicts the next frame of wildfire progression. Similar to  $\mathbf{x}_t$ , each pixel of  $\mathbf{z}_{t+1}$  can assume any value from the set S.  $\otimes$  demotes the convolution operation,  $\sigma$  denotes the sigmoid activation function, and t anh is the hyperbolic tangent function. The variables  $i_t$ ,  $f_t$ , and  $o_t$  correspond to the input, forget, and output gates, respectively, which regulate the information flow within the memory cell. The term  $\tilde{C}_t$  represents the candidate cell state,  $C_t$  is the current cell state, and  $\mathbf{z}_t$  is the hidden state or output of the ConvLSTM cell. Convolutional kernels  $W_{xi}$ ,  $W_{xf}$ ,  $W_{xo}$ , and  $W_{xs}$  are applied to the input feature landscape  $\mathbf{x}_t$ , while kernels  $W_{hi}$ ,  $W_{hf}$ ,  $W_{ho}$ , and  $W_{hs}$  are applied to the previous hidden state  $\mathbf{z}_t$ . The bias terms  $b_i$ ,  $b_f$ ,  $b_o$ , and  $b_s$  are associated with the input, forget, output gates, and cell state candidate, respectively. For brevity, the prediction length is set to one time step in Equation (6).

Our ConvLSTM model is designed for a 3-to-3 prediction task, as outlined in Algorithm 1. To simplify processing, the burned area data are resized to 128×128 pixels. The model takes three consecutive 128×128 matrices as input, each representing a time step in the fire progression sequence. These matrices encode fire dynamics based on the state definitions of the ConvLSTM model.

The state definitions differ slightly from those in the CA model. Using this multi-class approach, we identify both burning cells and track the duration of temporary firebreaks. The states are defined as follows:

- State 0: Unburned cells

## Algorithm 1 ConvLSTM Model Training

```
1: Hyperparameters:
 2: Learning rate: \alpha = 3e^{-4}
 3: Number of iterations: Iter = 5
 4: Batch size: b = 16
 5: Sequence length: l = 3
 6: Channel set: S = \{0, 1, 2, ..., 12\}
 7: Size of each frame: N \times M = 128 \times 128
 8: ConvLSTM model with parameters \theta: f_{\theta}^{\text{ConvLSTM}}
 9: Training Procedure:
10: Initialize ConvLSTM model parameters \theta
11: iter \leftarrow 0
12: while iter 

Figure 4. Data generation and training pipeline

170 positioning the ConvLSTM model as performing a multi-class classification task at each pixel and time-step. The structure of our model is detailed in Table 1.

The input tensor has a shape of (b, 3, 2, N, M), where b represents the batch size and  $N \times M$  denotes the spatial dimensions of the input field. This structure indicates that we have 3 sequential frames and 2 channels. One channel encodes fire information (a matrix where 0 indicates that the pixel has not been affected by fire, 1 indicates that a pixel is either currently burning or has burned and other is for the firebreak states which is from 2 to 12). The second channel was originally designed to include land-scape data; however, integrating such detailed information requires a comprehensive dataset, which is not currently available. Consequently, this channel is zero-filled in the current implementation. Therefore, the current ConvLSTM implementation is tailored to the three landscapes used in the CA simulation, each corresponding to a distinct training dataset.

For the ConvLSTM layers, each unit maintains a hidden state and a current state, with 128 feature channels and a sequence length of 3. Thus, the ConvLSTM output tensor has a shape of (b, 3, 2, 128, N, M). After passing through the subsequent 3D

| Component   | Layer                                     | Output Shape                          | Activation    |  |
|-------------|-------------------------------------------|---------------------------------------|---------------|--|
| Encoder     |                                           |                                       |               |  |
| Input       | -                                         | (b, 3, 2, N, M)                       | -             |  |
| ConvLSTM    | 128 channels, $3 \times 3$ kernel         | (b, 3, 2, 128, N, M)                  | Sigmoid, Tanh |  |
| ConvLSTM    | 128 channels, $3 \times 3$ kernel         | $(b,3,2,128,\mathcal{N},\mathcal{M})$ | Sigmoid, Tanh |  |
| Decoder     |                                           |                                       |               |  |
| Input       | -                                         | (b, 3, 2, 128, N, M)                  | -             |  |
| ConvLSTM    | 128 channels, $3 \times 3$ kernel         | (b, 3, 2, 128, N, M)                  | Sigmoid, Tanh |  |
| ConvLSTM    | 128 channels, $3 \times 3$ kernel         | $(b,3,2,128,\mathcal{N},\mathcal{M})$ | Sigmoid, Tanh |  |
| Convolution |                                           |                                       |               |  |
| Input       | -                                         | (b, 3, 2, 128, N, M)                  | -             |  |
| Conv3d      | 16 channels, $1 \times 3 \times 3$ kernel | $(b,3,16,{\rm N,M})$                  | -             |  |

Table 1. ConvLSTM Model Summary

convolutional layers, the final output of the model has a shape of (b, 3, 16, N, M), where the sequence length is 3 (corresponding to 3 consecutive frames) and 16 represents the number of prediction categories (e.g., different fire spread states or fire suppression strategies).

| Dataset    | Firebreak Type | Firebreak Placement Time | # CA Simulations | # Snapshots |
|------------|----------------|--------------------------|------------------|-------------|
| Train      | 3T1P           | T: 2, 4, 6; P: 3         | 1000             | 26000       |
| Validation | 3T1P           | T: 2, 4, 6; P: 3         | 100              | 2600        |
| Test       | 3T1P           | T: 2, 4, 6; P: 3         | 100              | 2600        |
|            | 3T             | T: 2, 4, 6; P: —         | 100              | 2600        |
|            | 2P             | T: —; P: 3, 5            | 100              | 2600        |
|            | None           | T: —; P: —               | 100              | 2600        |

Table 2. Summary of dataset distribution and characteristics for training, validation, and testing for each model, categorized by landscapes: 'Bear 2020', 'Chimney 2016', and 'Ferguson 2018'. The column '# CA Simulation' indicates the number of CA simulations (with different fire ignitions) that generated the datasets, with each simulation producing '# Snapshots' representing the number of CA time-steps. The model's parameters are based on each landscape's local geological characteristics, including vegetation, slope, wind speed, and other factors. 'T' represents temporary firebreak placement times, and 'P' represents permanent firebreak placement times. For each simulation, the ignited field is randomly selected, and the firebreak is randomly positioned around the ignited field.

The ConvLSTM model is trained using Cross Entropy Loss on the training dataset for Chimney 2016', 'Bear 2020', and 'Ferguson 2018' fires (Table 2) and validated on a distinct validation dataset using three metrics: Mean Squared Error (MSE), Structural Similarity Index Measure (SSIM), and Relative Prediction Error (RPE).

The MSE measures the average squared difference between the predicted  $(\mathbf{z}_t)$  and observed  $(\mathbf{x}_t)$  values. This metric evaluates the overall prediction accuracy of the model, with a lower MSE value indicating better performance.

$$MSE(\mathbf{z}_t, \mathbf{x}_t) = \frac{1}{b} \sum_{t=1}^{b} ||\mathbf{x}_t - \mathbf{z}_t||^2$$
(8)

The SSIM evaluates the structural similarity between the predicted and true images by considering luminance, contrast, and structural information. It ranges from -1 to 1, where a value close to 1 indicates a high degree of similarity.

$$SSIM(\mathbf{z}_t, \mathbf{x}_t) = \frac{(2\mu_{\mathbf{x}_t}\mu_{\mathbf{z}_t} + c_1)(2s_{\mathbf{x}_t\mathbf{z}_t} + c_2)}{(\mu_{\mathbf{x}_t}^2 + \mu_{\mathbf{z}_t}^2 + c_1)(s_{\mathbf{x}_t}^2 + s_{\mathbf{z}_t}^2 + c_2)}$$
(9)

The RPE is defined as the ratio of mismatched pixels between the predicted and observed fire spread landscapes, relative to the total number of pixels  $(N \times M)$ . This metric provides an intuitive measure of the model's ability to correctly classify the fire spread.

$$RPE(\mathbf{z}_t, \mathbf{x}_t) = \frac{\#\{\mathbf{x}_t \neq \mathbf{z}_t\}}{N \times M}$$
(10)

where  $\#\{\mathbf{x}_t \neq \mathbf{z}_t\}$  represent the number of mismatched pixels.

Figure 5. Autoregressive fire spread predictions and compare to CA generated data.

To assess the predictive capabilities and computational performance of our model, several wildfire datasets with different configurations were used to set up four test scenarios: (a) Fire Propagation without Firebreaks: For this test scenario, a random ignition field was selected on the landscape, and fire propagation was simulated by using a CA simulator. (b) Fire propagation simulation with artificial firebreaks, where artificial firebreaks were randomly placed around a randomly chosen ignition point. The cases studied include the following configurations: two permanent firebreaks (2P), three temporary firebreaks (3T), and three temporary firebreaks combined with one permanent firebreak (3T1P). (c) Autoregressive fire spread predictions, to test the long-term predictive horizon and stability of the model in successive iterations. Given three time steps as initial input from the fire propagation simulation, the model generates the next three time steps. These outputs are then fed recursively back into the model as inputs for the next sequence. This process was repeated until the forecast extended to fifteen time steps beyond the initial input set (Fig. 5). (d) Comparative computational efficiency. To compare the computational efficiency of our model and CA approaches, a parallel analysis of execution time and resource consumption was performed at six different spatial resolutions:  $128 \times 128, 256 \times 256, 384 \times 384, 512 \times 512, 640 \times 640,$  and  $768 \times 768.$ 

For the three landscapes considered, each model was tested with data derived from four distinct scenarios (Table 2): (1) three temporary firebreaks combined with one permanent firebreak (3T1P), (2) three temporary firebreaks (3T), (3) two permanent firebreaks (2P), and (4) no firebreak (None). The testing data were generated using the CA simulator and matched the configurations used during model training.

To evaluate the computational efficiency of our three ConvLSTM models, we compared their efficiency with that of the CA model across varying landscape resolutions. A series of experiments was conducted using simulated wildfires, each running for 150 time steps on landscapes of increasing size (from  $128 \times 128$  to  $768 \times 768$ ). For each landscape resolution, the runtime to predict three consecutive time steps—corresponding to a single ConvLSTM inference—was recorded. To ensure stability and consistency in the results, the average runtime was computed over the full 150-step simulation period. The goal was to evaluate the models' speed and their ability to handle large-scale landscapes, which is essential for real-world wildfire datasets that require high-resolution simulations to capture intricate spatial details.

#### 3 Results

#### 3.1 Model Performance

Analysis of the Chimney 2016, Ferguson 2018 and Bear 2020 simulations show that all metrics converge to acceptable levels: the Cross Entropy Loss during training phases (Fig. 6(a)), remains below 0.02, the MSE (Fig. 6(b)), remains below 0.01, RPE (Fig. 6(c)) stabilizes around 25 % and SSIM for the validation phase (Fig. 6(d)) is consistently higher than 0.9. These results show that the model is not only learning effectively but is also achieving strong performance on unseen data. The consistent validation metrics suggest that the model generalizes well and is robust against over-fitting.

All models demonstrated comparable performance across different configurations (Fig. 7):. Scenarios with firebreaks (3T1P, 3T and 2P) had higher prediction accuracy than those without (None). This improved performance can be attributed to the controlled conditions provided by firebreaks, which limit fire spread, reduce system randomness, and enhance prediction re-

Figure 6. Metric values during the training process.

240

liability. The absence of constraints on fire spread in the scenarios with no firebreaks introduces greater randomness in the CA simulation. The two-permanent-firebreak (2P) scenario had higher uncertainty than the other firebreak scenarios because the 90 % suppression rate introduced additional stochastic factors. Although both the no firebreak (None) and the two permanent firebreaks (2P) scenarios had poorer performance and higher standard deviations, nevertheless the results remain within acceptable limits.

Although iterative testing inevitably results in error accumulation over time, the models retained strong predictive accuracy even after multiple iterations. The average MSE was below 0.01 for the first three time steps, and remained under 0.03 across the entire 15-step sequence (Fig. 7(a), 7(d), 7(g)). The mean RPE was under 0.75 % for the first nine time steps, and stayed below 1.5 % for the full sequence (Fig. 7(b), 7(e), 7(h)). The mean SSIM value exceeded 0.99 for the first six time steps and remained above 0.97 throughout the entire sequence (Fig. 7(c), 7(f), 7(i)), indicating strong structural similarity between the predicted and actual fire spread.

Although there is an inherent challenge of cumulative errors in iterative modelling, the error maps (Fig. 8, B1, B2, B3), show only small deviations between the predicted and target values even after five loops. Thus, the models effectively maintain accuracy under iterative testing conditions and have successfully learned the dynamics associated with both temporary and

**Figure 7.** Testing models on metric MSE, RPE and SSIM with different configurations. The solid line is the mean of the data after filter outliers and the shadow represent the standard deviation.

permanent firebreaks. Specifically, the models recognize that temporary firebreaks disappear after ten time steps, achieving a suppression rate nearing 100 % while permanent firebreaks do not disappear but have a lower suppression rate. The CA model provides comparable accuracy to the DL model (Table 3). However, the DL model is significantly faster.

#### 3.2 Inference Speed Evaluation

The ConvLSTM model, executed on an NVIDIA A100 PCIE GPU, completed simulations of 3 time steps for all resolutions from 128×128 to 768×768 in under 0.2 seconds (Fig. 9). The ConvLSTM's computation time increased linearly with landscape size, whereas the CA model, executed on a CPU, showed significantly higher computational costs overall and computation time increased exponentially with larger landscape sizes. For the maximum resolution of 768 × 768, the CA model required nearly 50 seconds to predict three time steps, approximately 250 times longer than the ConvLSTM model. The faster speed of the ConvLSTM models arises partly from the efficiency of the model architecture but also from the advantages of GPU acceleration.

| Firebreak Type | Landscape                            | Model - | Metrics |        |        |
|----------------|--------------------------------------|---------|---------|--------|--------|
| Theoreak Type  |                                      |         | MSE ↓   | RPE ↓  | SSIM ↑ |
|                | D 2020                               | DL      | 0.0043  | 0.0037 | 0.9916 |
|                | Bear 2020                            | CA      | 0.0884  | 0.0046 | 0.9945 |
| 2T1D           | Chimney 2016                         | DL      | 0.0064  | 0.0036 | 0.9915 |
| 3T1P           |                                      | CA      | 0.0924  | 0.0050 | 0.9948 |
|                | Ferguson 2018                        | DL      | 0.0076  | 0.0053 | 0.9890 |
|                |                                      | CA      | 0.0780  | 0.0036 | 0.9952 |
|                | Bear 2020                            | DL      | 0.0061  | 0.0050 | 0.9885 |
|                |                                      | CA      | 0.0633  | 0.0025 | 0.9993 |
| 275            | Chimney 2016                         | DL      | 0.0064  | 0.0034 | 0.9925 |
| 3T             |                                      | CA      | 0.0923  | 0.0049 | 0.9943 |
|                | Ferguson 2018                        | DL      | 0.0075  | 0.0044 | 0.9908 |
|                |                                      | CA      | 0.0785  | 0.0043 | 0.9956 |
|                | Bear 2020                            | DL      | 0.0022  | 0.0022 | 0.9941 |
|                |                                      | CA      | 0.3633  | 0.0040 | 0.9906 |
| an.            | Chimney 2016<br>Ferguson 2018        | DL      | 0.0037  | 0.0037 | 0.9914 |
| 2P             |                                      | CA      | 0.2428  | 0.0016 | 0.9960 |
|                |                                      | DL      | 0.0027  | 0.0027 | 0.9931 |
|                |                                      | CA      | 0.1456  | 0.0011 | 0.9969 |
|                | Bear 2020 Chimney 2016 Ferguson 2018 | DL      | 0.0030  | 0.0029 | 0.9919 |
|                |                                      | CA      | 0.9947  | 0.0108 | 0.9804 |
| N              |                                      | DL      | 0.0055  | 0.0055 | 0.9876 |
| None           |                                      | CA      | 0.9897  | 0.0071 | 0.9840 |
|                |                                      | DL      | 0.0021  | 0.0020 | 0.9939 |
|                |                                      | CA      | 0.9499  | 0.0101 | 0.9804 |

**Table 3.** Comparison of Metrics Across Different Fire Suppression Strategies, Landscapes, and Models. For the DL model, the values represent the mean of the metric of the middle three time-steps' (sixth, seventh, eighth) tested on 100 CA simulations each has 26 CA time-steps (Table 2). For the CA model, the values represent the mean metric calculated over three time steps, using the same initial conditions across four separate simulations.

# 4 Conclusion and Future directions

The ConvLSTM model has good performance in simulating wildfire propagation combined with firebreak deployment. It accurately identifies different types of firebreaks, evaluates their efficiency, and predicts how long they last. In comparison with the CA model, which is simple and highly interpretable but has a high computation cost, the speed of ConvLSTM model across varying input sizes makes it useful for real-world, real-time applications.

**Figure 8.** Here we tested each landscape using a autoregressive testing approach, performing 5 iterative loops that will generate the following 15 time-steps and we plot the result for every 2 time-steps. For the error map, red means false negatives and blue means false positive.

Wildfire propagation is inherently stochastic, and the autoregressive approach amplifies prediction errors over time. The task becomes more difficult in high-altitude regions where burning is less likely, especially since explicit landscape features

270

275

Figure 9. Prediction time (in seconds) for completing 3 steps of simulation across various landscape resolutions ( $128 \times 128$  to  $768 \times 768$ ). The CA Model using CPU and the ConvLSTM Model using NVIDIA A100 PCIE GPU.

are not provided as model inputs. The absence of key drivers like wind speed and direction further limits predictive accuracy. Despite these challenges, the model performs well, accurately capturing wildfire spread and firebreak behaviour across diverse scenarios.

The model could be further improved by using detailed landscape data, including vegetation types, densities, moisture levels, and topographical features, as model inputs. This would allow the development of a universal model that performs well across various geographic regions, minimizing the need to tune the model for each fire event. Incorporating dynamic meteorological information and wind patterns would also enhance prediction accuracy. More realistic characterisation of firebreak placement and experiments to optimise placement under realistic conditions would also enhance the usefulness of the model and contribute to more context-aware forecasting, benefiting wildfire management and containment strategies.

Code and data availability. The code and data used in this study are publicly available in a GitHub repository and have been archived on Zenodo (Zheng, 2025). These resources are accessible under the terms of the MIT licence, which permits free use, modification and redistribution. Data on landscape slope, vegetation density, and vegetation cover for each ecoregion are obtained from IFTDSS (Drury et al., 2016).

### Appendix A: Acronyms

ML Machine Learning

RNN Recurrent Neural Network
LSTM Long Short-Term Memory

DL Deep Learning

ConvLSTM Convolutional Long Short-Term Memory

CA Cellular Automata

DEVS Discrete Event System Specification

MODIS Moderate Resolution Imaging Spectroradiometer

VIIRS Visible Infrared Imaging Radiometer Suite

IFTDSS Interagency Fuel Treatment Decision Support System

MSE Mean Squared Error

SSIM Structural Similarity Index Measure

RPE Relative Prediction Error

# Appendix B: More test example

**Figure B1.** Bear 2020 test example. In Fig. (a), (b), we performed 5 iterative loops, generating 15 time-steps, with the results plotted every 2 time-steps. In contrast, in Fig. (c), we conducted 7 iterative loops, generating 21 time-steps, and plotted the results every 3 time-steps. For the error image, red indicates false negatives, while blue represents false positives.

**Figure B2.** Chimney 2016 test example. In Fig. (a), (b), we performed 5 iterative loops, generating 15 time-steps, with the results plotted every 2 time-steps. In contrast, in Fig. (c), we conducted 7 iterative loops, generating 21 time-steps, and plotted the results every 3 time-steps. For the error image, red indicates false negatives, while blue represents false positives.

**Figure B3.** Ferguson 2018 test example. In Fig. (a), (b), we performed 5 iterative loops, generating 15 time-steps, with the results plotted every 2 time-steps. In contrast, in Fig. (c), we conducted 7 iterative loops, generating 21 time-steps, and plotted the results every 3 time-steps. For the error image, red indicates false negatives, while blue represents false positives.

Author contributions. JZ prepared the data, implemented the models, analyzed the results, and prepared the draft manuscript with the contributions of all co-authors. SC, SH and ZX brought domain expertise to the project, contributing to both result analysis and manuscript development. SC developed the research idea and provided the expertise for model choice and implementation. RA, SH, ZX, LX and SC reviewed and edited the manuscript.

Competing interests. The contact author has declared that none of the authors has any competing interests.

*Acknowledgements.* Sibo Cheng acknowledges the support of the French Agence Nationale de la Recherche (ANR) under reference ANR-25-CE56-0198. CEREA is a member of Institut Pierre-Simon Laplace (IPSL).

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
