# Peer review of "Predicting spatio-temporal wildfire propagation with dynamic firebreaks"

_EGUsphere, 2025_

## Referee Comment (RC1)

**Predicting spatio-temporal wildfire propagation with dynamic firebreaks**

**egusphere-2025-4007**

**General Comments**

This is a well-structured research paper on introducing fire breaks to simulations that predict fire spread, while detailing the different machine learning algorithms that take this important element into account. While the research is novel and the methodology is reproducible, the impact it may have in actual wildfire situations is questionable. The approach that the authors propose can certainly be used to better test/understand wildfire dynamics in virtual environments, and may even be used to train first responders, however it falls short of becoming a core application in case of a live wildfire mainly for two reasons: 1- Wildfire propagation is greatly related to wind, and with climate change we see higher uncertainty and extremes in wind patterns. Coarse resolution wind data may grossly underestimate what is going on in the actual wildfire scene, as not only wind speed, but wind gusts during an active fire are major players in fire severity, and spread (starting new ignitions in forest patches that are far away from the main ignition zone, regardless of a fire break). Also, a severe wildfire can produce its own wind patterns, shift the current wind direction, or increase its intensity. The authors state that they are aware of this shortcoming in their study. But this undermines their claim that their methodology may prove as an effective strategy to reduce wildfire impacts, since wind can also affect how temporary breaks are (or can be) deployed. 2- Wildfire propagation is also greatly related to topography, but the authors mention that they did not have a comprehensive landscape dataset available, so the ConvLSTM simulations were run with null input for landscape.

As a learning and potential training tool, despite its limitations, I find the work insightful with room for improvement, a possible first step towards developing a global dynamic simulator that considers fire breaks while projecting potential wildfire spread and provide valuable insight for effective containment. The clear methodology helps the simulations' reproducibility, and aid researchers to test it in their study areas. However, speed vs accuracy between CA and DL models needs to be carefully considered, as one should not replace the other.

major revision

**Technical Comments**

There is little to no discussion of the simulation results.

All figures containing spatial information (maps) need improvement:

In Figure 1 the color selection makes it hard to interpret figures b, c, f and g. The natural color figures are also too small to see, "e" is of different size.

In Figure 3 the simulation results are hard to see over a colorful background (especially when we are not sure what the colors denote, is it land cover?), either crop to the simulation zone, or utilize a blow-up window to show us the simulation results separately and in close up fashion. Alternatively, if you will not

make any reference to the background, you can neutralize it with a filter, or other color selection so the BA and fire breaks are more visible, and preferably larger (same for Figure 8).

There is a Figure 4, but there is no reference to it in the text. I would have preferred it was introduced towards the beginning of the methodology when the reader is trying to visualize how the experiment runs. May be a few sentences before the detailed explanations of CA and ConvLSTM models introducing the workflow, and referencing the figure would also help understand the grid structure better, through actual visualization.

In Section 3.2 the authors compare the speed of a CPU run simulation to a GPU run simulation, which will produce slower results. To be able to compare both simulations head on, they should be run on the same set up. CA model may run slower but from the rate quoted here it is unclear how much of it comes from the machine how much of it from the model's own performance. Therefore a "250x" expression should be re-evaluated.

In the simulation results shown in Figure 8, there is a varying degree of false negative and positives among the three text examples. Ferguson fire being the smallest whereas Chimney Fire showing several (Bear Fire also). It is expected for a model that is trained on model data to exponentially over/under predict overtime, however the difference among the three test cases could have been better explained in the text. Also, I would expect to see a ratio timeseries (along with the map demarcations) so it is easier to interpret accurately. In an attempt to explain the false positive/negatives, a mention of landscape data limitation is mentioned here and wind speed, but the reader would appreciate a more in-depth explanation/discussion. Also "Despite these challenges, the model performs well, ..." is a bit of an overstatement given the results, toning that down may help meet expectations.

In sum, the authors undertake an important task, including fire breaks (and their efficiency) in fire propagation simulations. Among the pros the work's easy reproducibility tops the list due to a clear methodological break down. However, these series of experiments are limited in capacity since they fall short of considering wind (speed and direction) as well as topography. The manuscript has room for improvement, especially through a solid discussion of results.